# CONFIDENCE-BASED MODEL SELECTION: WHEN TO TAKE SHORTCUTS IN SPURIOUS SETTINGS

## ABSTRACT

Effective machine learning models learn both robust features that directly determine the outcome of interest (e.g., an object with wheels is more likely to be a car), and shortcut features (e.g., an object on a road is more likely to be a car). The prevailing sentiment in the robustness literature is to avoid such correlative shortcut features and learn robust predictors. However, while robust predictors perform better on worst-case distributional shifts, they often sacrifice accuracy on majority subpopulations. In this paper, we argue that shortcut features should not be entirely discarded. Instead, if we can identify the subpopulation to which an input belongs, we can adaptively choose among models with different strengths to achieve high performance on both majority and minority subpopulations. We propose COnfidence-baSed MOdel Selection (CoSMoS), where we observe that model confidence can effectively guide model selection. Notably, CoSMoS does not require any target labels or group annotations, either of which may be difficult to obtain or unavailable. We evaluate CoSMoS on four datasets with spurious correlations, each with multiple test sets with varying levels of distribution shift. We find that CoSMoS achieves 2-5% lower average regret across all subpopulations, compared to using only robust predictors or other model aggregation methods.

## 1 INTRODUCTION

Datasets often exhibit spurious correlations, where a classifier based on a *shortcut feature* that is predictive on the training data can be misled when faced with a distribution shift in inputs (Geirhos et al., 2020). For example, consider the task of classifying cows or camels, where most cows in the source distribution have grass backgrounds while most camels have sand backgrounds. Standard models trained with empirical risk minimization (ERM) optimize for average performance and may learn a predictor that relies on the background of the image for this task. This reliance on shortcut features can result in subpar performance when the model is tested on data distributions with a larger representation from regions of minority subpopulations.

The robustness literature has traditionally entirely discarded shortcut features, and the term carries a negative connotation. In particular, recent works have proposed various debiasing methods to counteract the issues arising from shortcut features (Sagawa et al., 2020; Nam et al., 2020; Liu et al., 2021). The aim of these debiasing methods is to learn an "invariant predictor"—a function that is invariant to changes in features that bear no causal relationship with the label. However, these methods often result in lower average accuracy compared to models that use shortcut features, as they necessarily sacrifice accuracy on majority subpopulations. In the cow/camel task, while we do want our predictors to focus on the animal, an invariant predictor may entirely discard the background information. This information can still be valuable, particularly since the animal itself may be difficult to classify in some images.

As they both have strengths and weaknesses on different subpopulations, we argue for *viewing shortcut and invariant classifiers on equal footing, where both are experts but on different regions of the input space*. This shift in mindset may lead to a more optimal strategy for these distribution shifts – provided we can discern when to include or exclude these features in our predictions, we can exploit the advantages of both types of models in a complementary way. Prior work has studied this phenomenon in human decision-making (Tversky & Kahneman, 1974; Simon et al., 1989; Gigerenzer & Gaissmaier, 2011): sometimes, adopting informal mental shortcuts is more effective than making a complete logical analysis based on all available information. In this paper, we demonstrate

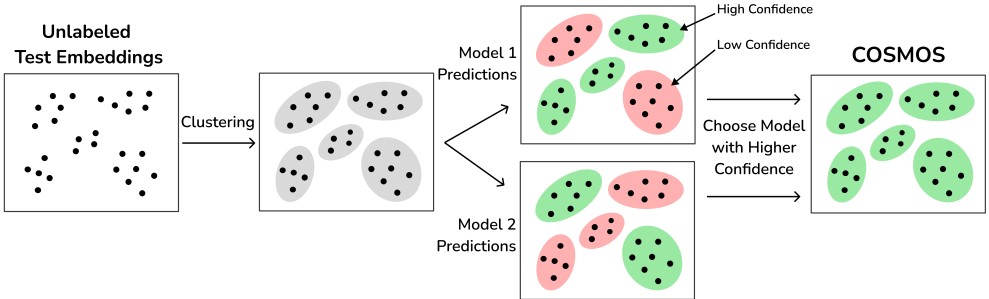

Figure 1: **Confidence-based Model Selection (CoSMoS).** We start with multiple base models with diverse performance characteristics. After clustering test embeddings, CoSMoS routes each cluster to the base model that has highest average predictive confidence for that cluster. The final aggregated classifier leverages the strengths of each model, achieving higher performance.

the benefits of leveraging the strengths of both shortcut and debiased classifiers in achieving high performance on both majority and minority subpopulations.

Instead of combining models as done in typical ensemble methods, we use different models for different inputs. We hypothesize that selectively employing an appropriate classifier for different inputs maintains high performance more effectively on both majority and minority subpopulations compared to relying on a single predictor. In practice, we find that simply using model confidence can effectively determine which model to choose for different inputs, and we propose COnfidence-baSed MOdel Selection (CoSMoS) which, given multiple base classifiers, selectively employs an appropriate classifier for each input in the test set. Since model confidence can be noisy on a single data point, we propose clustering inputs before model assignment as a means of variance reduction, using average model confidence on that cluster to determine which classifier to use. Our approach is general in that it may be applied to any set of base classifiers, including ones in different loss basins, and it does not require access to the weights of the base classifiers or to any group annotations. Furthermore, many prior works that study robustness to spurious correlations subtly require access to labeled target distribution data when tuning hyperparameters, whereas our approach does not require any access to any labeled data from the target domain. A summary is given in Figure 1.

We evaluate our method on four datasets with spurious correlations, each with many test sets with various levels of data distribution shift. We consider many test sets because when using models in the real world, we do not know which test set the model will be evaluated on. Thus, we want models to perform reliably on a wide range of subpopulations, including those that are over- and under-represented during training. In other words, we aim to satisfy multiple desirable objectives– achieving high accuracy on unseen data from both the source distribution and the worst-group target distributions. Our results show that our method achieves 2-5% lower average regret across the subgroups of the input space compared to using a single predictor or methods that aggregate multiple classifiers. In particular, our approach can achieve high accuracy on minority subpopulations without sacrificing performance on majority subpopulations. Furthermore, we show that our method can also be used for other tasks where we need to choose the best model for a test distribution. For example, by taking the candidate that is chosen the most by CoSMoS, we can use CoSMoS to do hyperparameter tuning without any labels from the desired test distribution, and to our knowledge, we are the first method to do so.

## 2 RELATED WORK

**Robustness and Adaptation Using Unlabeled Target Data.** Many prior works aim to improve robustness to various distribution shifts (Tzeng et al., 2014; Ganin et al., 2016; Arjovsky et al., 2019; Sagawa et al., 2020; Nam et al., 2020; Creager et al., 2021; Liu et al., 2021; Yao et al., 2022; Zhang & Ré, 2022; Yao et al., 2023). Additionally, prior works have studied how to adapt pre-trained features to a target distribution via fine-tuning Oquab et al. (2014); Yosinski et al. (2014); Sharif Razavian et al. (2014). Such fine-tuning works frame robustness to distribution shift as a zero-shot generalization problem Kornblith et al. (2018); Zhai et al. (2019); Wortsman et al. (2022b); Kumar et al. (2022), where the model is fine-tuned on source data and evaluated on data from the target distribution. Several recent works have also studied how to adapt to a target distribution using unlabeled target data at test time Lee et al. (2013); Ganin et al. (2016); Wang et al. (2020); Zhang

et al. (2022). In contrast to these single-model methods, this paper presents a simple and novel approach that capitalizes on multiple models to address the distribution shift caused by spurious correlations. The model selection strategy of COSMOS is orthogonal to the specific methods above, and can in principle leverage models trained with any of those techniques.

**Diverse Classifiers.** Neural networks, by their nature, often exhibit a bias towards learning simple functions that rely on shortcut features (Arpit et al., 2017; Gunasekar et al., 2018; Shah et al., 2020; Geirhos et al., 2020; Pezeshki et al., 2021; Li et al., 2022; Lubana et al., 2022). To better handle novel distributions, prior works consider the entire set of functions that are predictive on the training data (Fisher et al., 2019; Semenova et al., 2019; Xu et al., 2022). Recent diversification methods show how to discover such a set (Teney et al., 2022; Pagliardini et al., 2022; Lee et al., 2022b; Chen et al., 2023) and show that there are multiple predictors that perform well on the source domain but differently on different test domains. COSMOS assumes access to a diverse set of classifiers that were trained with different strategies, with a focus on the selection stage. We choose an appropriate classifier for each input region in an unsupervised way, whereas most existing works for choosing among diverse classifiers do a form of adaptation using labels.

**Estimating Confidence.** Prior works have studied various metrics for estimating model confidence, such as softmax probability (Pearce et al., 2021), ensemble uncertainty (Lakshminarayanan et al., 2017), or prediction entropy (Pereyra et al., 2017). These metrics are often studied in the context of uncertainty quantification for out-of-distribution detection (Hendrycks & Gimpel, 2016; Lee et al., 2018; Ovadia et al., 2019; Berger et al., 2021). COSMOS first calibrates each classifier via temperature scaling, and then uses the softmax probability as an estimate of the confidence of each model. Future advancements in calibration and confidence estimation methods Guo et al. (2017); Liang et al. (2017) can be incorporated into the COSMOS framework to further improve performance.

**Model Selection and Ensemble Methods.** Ensemble methods improve performance by integrating predictions from several models (Dietterich, 2000; Bauer & Kohavi, 1999; Hastie et al., 2009; Lakshminarayanan et al., 2017; Izmailov et al., 2018; Wortsman et al., 2022a). Recognizing that different models may be best suited for different inputs, this paper investigates optimal model selection in an input-specific fashion. Previous works aggregate the information in different models through feature selection (Dash & Liu, 1997; Liu & Motoda, 2007; Chandrashekar & Sahin, 2014; Li et al., 2017) or model selection by estimating the accuracy of each model on unlabeled target data (Garg et al., 2022; Baek et al., 2022). Our method is an example of dynamic selection (DS) from multiple classifiers (Cruz et al., 2018; Sellmann & Shah, 2022; Cruz et al., 2020; Ekanayake et al., 2023); we propose a distinct selection criteria, combining both clustering and confidence strategies to enhance performance on subpopulation shifts. Finally, there is an extensive body of work on mixture of experts (MoE) methods, which train a set of specialized experts and a mechanism to route datapoints to experts (Jacobs et al., 1991; Jordan & Jacobs, 1994; Yuksel et al., 2012; Masoudnia & Ebrahimpour, 2014). While COSMOS is similar in spirit to MoE methods, it is much simpler and does not require any additional training.

## 3 PROBLEM SETTING: SPURIOUS CORRELATIONS

We now describe our problem setting, where the goal is to provide an accurate decision boundary under a wide range of target distributions for datasets with spurious correlations. In spurious settings, the source and target distributions are mixtures of a common set of subpopulations, but the relative proportions of these subpopulations may differ. Because of the different subpopulation proportions, certain groups may be over- or under-represented in the (source) training data. Importantly, we do not assume access to target labels, or any form of prior knowledge of the target distributions. We also do not assume access to group labels of any kind; i.e. we do not know the subpopulation that a datapoint belongs to, nor do we even know the number of underlying groups.

More formally, we consider a source distribution $p_S(x, y)$ and multiple target distributions $p_T^1(x, y), p_T^2(x, y), \ldots$, each corresponding to a different distribution over subpopulations. The source dataset $\mathcal{D}_S \in (\mathcal{X} \times \mathcal{Y})^N$ is sampled according to the source distribution $p_S$. We evaluate performance on each target distribution $p_T^i$ on test datapoints $\mathcal{D}_T^i$. Let $z$ be a hidden index variable that indicates the subpopulation of a datapoint. We assume that for every target distribution $T_i$, the following invariances hold: $p_S(y) = p_{T_i}(y)$, $p_S(y|x, z) = p_T^i(y|x, z)$, and $p_S(x|y, z) = p_T^i(x|y, z)$. The main difference across distributions is the relative

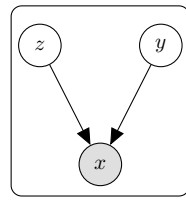

proportions of subpopulations $p(z)$. Our goal is to find the optimal predictor $p_T^i(y|x_i)$ for any desired subpopulation $p_T^i$.

We assume access to diverse classifiers $f_1, f_2, \ldots, f_K$ that are trained on the source data $\mathcal{D}_S$, potentially using different objectives or pre-trained backbones, etc. Classifiers trained in different ways are known to have different inductive biases, making different classifiers perform better on different subpopulations. In other words, if we consider two subpopulations denoted $z = 0$ or $z = 1$, a classifier $f_i$ may achieve higher accuracy than another $f_j$ on $p(y|x, z = 0)$, while a different classifier may be best on $p(y|x, z = 1)$. We do not make any specific assumptions on the training procedure of each classifier aside from leveraging the fact that they have complementary strengths.

We note that our setting differs from the setting studied in some prior works on spurious correlations (Sagawa et al., 2020), which evaluate the model's performance only on the hardest target distribution (i.e., worst-group accuracy). Our setting also differs from those that use a small amount of target data to fine-tune (Lee et al., 2022a; Chen et al., 2023). It may be expensive to acquire any target labels, and we desire a method that can be applied to any target distribution without requiring additonal information from the domain.

## 4 CONFIDENCE-BASED MODEL SELECTION (COSMOS)

In this section, we describe COSMOS, a framework for leveraging multiple models that adaptively chooses an appropriate model for each input. In the previous section, we described our problem setting, where we consider spurious settings, where there exist shifts in the frequencies of data subpopulations. There are multiple good classifiers that perform well on $p_S(x, y)$ but may perform differently on different subpopulations determined by $z$ and therefore also on unknown test distributions $p_{T_i}(x, y)$. Hence, to leverage the strengths of these classifiers, if we can determine which $z$ each test input belongs to, we can just choose the corresponding classifier that performs best for that subpopulation. However, we do not have any labels on $z$ and do not even know how many different groups there are. To avoid needing to explicitly classify the $z$ for each datapoint, we instead estimate which datapoints have similar $z$ and use confidences of the classifiers to implicitly indicate a subpopulation identity.

COSMOS is based on the intuition that a well-calibrated classifier can indicate its suitability for a given input. The classifier with the highest confidence on an input is likely the best choice for the subpopulation $z$ which contains that input. We acknowledge that calibration is a difficult problem but find that simple temperature scaling suffices for our experiments. If all classifiers are perfectly calibrated for each test input, we can solely rely on the predictive confidence, as indicated by softmax probabilities, to select the best classifier for each input. However, due to mismatches between training and test distributions, classifiers may not be perfectly calibrated on test data, so the confidence estimates may be noisy. If the inputs are mapped to a embedding space where inputs from the same population are closer together, we can mitigate the variance in these estimates by first clustering the test inputs, as one cluster will likely contain inputs from the same subpopulation. Thus, as long as it successfully groups inputs from the same population together, clustering may give a clearer signal of which implicit $z$ an input belongs to, and we should choose the appropriate classifier according to which has the the highest *average* confidence on each cluster of inputs. We find that this approach effectively reduces noise in the confidence estimates through a smoothing effect.

### 4.1 FORMAL INTUITION

We now provide some more formal intuition on our method. While we make some stronger assumptions in this section, we do not strictly require these assumptions in practice. The purpose of the assumptions here is to provide clear intuition for why our method works by establishing a connection to entropy minimization. Consider the graphical model presented in Section 3, where $p(x, z, y)$ factorizes as $p(x|z, y)p(z)p(y)$. Assume each model $f_i$ in our class of base classifiers $\{f_1, f_2, \ldots, f_K\}$ corresponds to some subpopulation $z_i$ on which the model performs the best among the set of models. Let us denote the set $\{z_1, \ldots, z_k\}$ as $\mathcal{Z}_k$. Now, if we restrict the support of $p(z \mid x)$ from our graphical model above to the set $\mathcal{Z}_k$, then if we have knowledge of $p(z \mid x)$ on every datapoint, we can then choose to use the model (from our class) that has the highest probability of being correct on that datapoint. Consequently, given some unlabeled data points from the target distribution $x_1, x_2, \ldots, x_m$, we attempt to solve an inference problem over $p(z \mid x)$. But, issues arise because we are only given unlabeled data from the target domain, and without any knowledge of true la-

bels $y$ on the set of target datapoints $x_1, ..., x_m$, or access to a hold out set with information on $z$ (group/subpopulation information), the problem remains a bit ill-posed.

One way to make the problem more identifiable, is to first estimate the label $\hat{y}_i$, on each test point $x_i$, and then solve the maximum likelihood estimation problem:

$$\underset{\{p(z|x_i)\}_{i=1}^m}{\arg\max} \sum_{i=1}^m \log p(\hat{y}_i \mid x_i) = \underset{\{p(z|x_i)\}_{i=1}^m}{\arg\max} \sum_{i=1}^m \log \int_z p(\hat{y}_i \mid x_i, z) \cdot p(z \mid x_i) \, \mathrm{d}z$$

Now, under the assumption that $p(z \mid x)$ is smooth in some metric (, for two data points $x_1, x_2$ that are close to each other in some metric space, $p(z \mid x_1) \approx p(z \mid x_2)$), we can solve the above inference problem after we cluster $x_1, x_2, \ldots, x_m$ into disjoint clusters under the metric space over which the smoothness assumption holds. If all data points in the cluster share the same $p(z \mid x)$, we can write the solution of the above likelihood maximization problem as:

$$p(z \mid x_i) = \delta_{z(x_i)} \text{ where, } z(x_i) = \arg\max_z \sum_{x' \in C(x_i)} p(\hat{y}_i \mid z, x'),$$

where $\delta_z$ is a Dirac delta function on some point $z$ in our set $\mathcal{Z}_k$ and $C(x_i)$ is the cluster assignment of the point $x_i$. The better the pseudolabels are, the more the assignment $z(x_i)$ corresponds to a specific subpopulation in set $\mathcal{Z}_k$.

Additionally, prior works have shown in different problem settings how pseudolabeling can be motivated through entropy minimization (Chen et al., 2020). For our setting, if the calibration of each model gives a reliable signal, and for each subpopulation we expect one model in particular to be the best, then the model assignment that achieves the lowest entropy will achieve highest performance. Furthermore, the predictive entropy of single instances can have high variance, and we can mitigate this by the smoothed estimate of average confidence within each cluster. Hence, we can also interpret the inference problem as smoothed entropy minimization, with the following objective:

$$\min_{z_i} \sum_i H(p(y|x, z_i)) + \sum_{j,k} \mathrm{KL}(p(z_i|x_j) || p(z_i|x_k)) \lambda_{jk},$$

where $\lambda_{jk} \to \infty$ for points in the same cluster and 0 otherwise. Our method can therefore be derived as performing MLE estimation on top of pseudo-labels, which is equivalent to minimizing the entropy of predictions with a smoothing regularizer term. This motivates our use of confidence in classifier selection, as models with low entropy correspond to models with high confidence.

### 4.2 Practical Method

We now describe our practical method in detail. We are given base classifiers $f_1, f_2, \ldots, f_K$ that are trained using the source data $\mathcal{D}_S$. We start with a simple calibration procedure and then cluster the test inputs so that we can select one of the base classifiers to use for each cluster of examples.

**Calibration.** We will use the softmax probability as our measure of confidence, so each model needs to first be calibrated so that the probabilities outputted for different inputs match the expected accuracy on those inputs. We can use the Expected Calibration Error (ECE) (De-Groot & Fienberg, 1983; Naeini et al., 2015) to calibrate each model, with the following procedure: We calculate the ECE on held-out data from the source distribution using the model's logits temperature scaled by different values of $\alpha$. Formally, $\mathrm{ECE}(f_i) = \sum_{j=1}^{10} P(j) \cdot |o_j - e_j|$, where $o_j$ is the true fraction of positive instances

---

**Algorithm 1** Confidence-Based Model Selection

1: **Input:** $\{f_1, ..., f_k\}$ base classifiers.
2: **for** each classifier $f_i$ **do**
3:     Calibrate with ECE to obtain temperature $\alpha_i$.
4: **for** test input $x$ **do**
5:     **for** each classifier $f_i$ **do**
6:         Compute confidence $C(x, f_i) = \arg\max_y p(y|f_i(x)/\alpha_i)$
7: Cluster test embeddings with K-means into $c_1, c_2, ..., c_k$.
8: **for** test input $x$ in each cluster $c_j$ **do**
9:     Select classifier $f^* = \arg\max_{f_i} \frac{1}{N} \sum_{x_j \in c_j} C(x_j, f_i)$.
10:     Predict label $f^*(x)$
11: **return** predicted labels

---

in bin $j$, $e_j$ is the mean of the post-calibrated probabilities for the instances in bin $j$, and $P(j)$ is the fraction of all instances that fall into bin $j$. Let $\alpha_i^*$ be the $\alpha$ that gives the lowest ECE $e_i^*$ for classifier $f_i$. We take the model with the highest $e^*$ and then choose $\alpha_i$ for every other model that gives an ECE closest to that $e^*$.

**Selecting a classifier for an input.** Now we consider our test set, for which we do not have any labels. We calculate the test logits scaled by the chosen $\alpha_i$ for each classifier. For each input $x$ in the test set and each classifier $f_i$, we calculate the confidence of that classifier's prediction as the softmax probability, i.e. $C(x, f_i) = \arg\max_y p(y|f_i(x)/\alpha_i)$. Our goal is to find the most appropriate classifier for each input. However, using the softmax probability as the confidence for each classifier can be noisy for individual inputs. Prior works have shown that clustering can recover meaningful subpopulations with various correlations (Sohoni et al., 2020). Thus, as long as the embedding space maps inputs from the same population closer together, clustering can alleviate the noise in input-level confidence estimates. Given a test set, we cluster the logits using K-means into $k^i$ clusters $c_1, c_2, ..., c_{k^i}$, where $k = \frac{|\mathcal{D}_T^i|}{N}$. We use $N = 50$ and do not tune $N$ in our experiments but do show the effect of using different values in Section 5.5. For each cluster, we then select the classifier with the highest average confidence on the points in the cluster, i.e. $f^* = \arg\max_{f_i} \frac{1}{N} \sum_{x_k \in c_j} C(x_k, f_i)$. In some cases, the embedding space may not be learned well enough so that clustering may not be able to recover the subpopulations, so we also a consider a variant of our method that does not use clustering, which we call CoSMoS (input-dep). We provide a summary of CoSMoS (cluster) in Algorithm 1.

Our method is designed to benefit particularly when given shifts in the frequencies of data subpopulations, as different classifiers often perform better on different subpopulations in this setting. However, the method can also be used to generalize to novel data subpopulations that were not observed during training. Our method is designed to be general to best maximize average accuracy using any given combination of classifiers for any test subpopulation. Thus, as we show in Sec. 5, we can use our method to do tasks like hyperparameter tuning. Importantly, CoSMoS does not require any exposure to labeled target data, any group annotations, or access to model weights.

## 5 EXPERIMENTS

In this section, we seek to empirically answer the following questions: (1) Can CoSMoS achieve higher average accuracy than using individual classifiers or other model aggregation methods on a wide range of subpopulations without any labeled data from the target domain? (2) Can we use CoSMoS for other use cases such as hyperparameter tuning? (3) How sensitive is CoSMoS to design choices such as cluster size? Below, we first describe our experimental setup, including our datasets, base classifiers, and evaluation metrics, along with experiments to answer the above questions.

### 5.1 EXPERIMENTAL SETUP

**Datasets.** We run experiments on the following four datasets: (1) **Waterbirds** (Sagawa et al., 2020), (2) **CelebA** (Liu et al., 2015), (3) **MultiNLI** (Williams et al., 2018), and (4) **MetaShift** (Liang & Zou, 2022). We provide descriptions of each dataset in Appendix A. For all settings, the base classifiers are trained on the original source datasets, and we use held-out data from the source distribution in order to calibrate the models. For each dataset, we construct multiple target distributions for evaluation representative of a range of potential test distributions, consisting of different subpopulations from either (a) mixing majority and minority groups or (b) each individual group. More specifically, we evaluate on the following subsets of the original test dataset: subsets where majority samples make up $m \in \{0, 10, 30, 50, 70, 90, 100\}$ percent of the samples with minority samples constituting the remaining samples, as well as each of the individual groups.

**Base Models.** We use base classifiers trained using ERM, JTT (Liu et al., 2021), and LISA (Yao et al., 2022) using their released codebases. For the image datasets, we use a ResNet-50 backbone pre-trained on ImageNet, and we use a pre-trained BERT model for MultiNLI. We take the hyperparameters described in the respective papers, although we show in Sec. 5.4 that we can actually do hyperparameter tuning with CoSMoS. We do not have access to any target domain labels. We calibrate each model using $\alpha$ between 0.25 and 15 in 0.25 increments. CoSMoS uses cluster size $k = \frac{|\mathcal{D}_T^i|}{N}$ for each target dataset $\mathcal{D}_T^i$, but we do not tune $N$ and instead just use $N = 50$ for each dataset.

| | Majority Groups | | Minority Groups | | Avg Acc | WG Acc |
|---|---|---|---|---|---|---|
| | LB+L | WB+W | LB+W | WB+L | | |
| Robust Classifier (JTT) (Liu et al., 2021) | 92.99 (0.06) | 95.36 (0.39) | **86.77 (1.46)** | **87.55 (0.08)** | 94.9 | **86.77** |
| Shortcut Classifier (ERM) | **96.74 (0.06)** | **99.24 (0.21)** | 76.09 (1.20) | 76.82 (0.08) | **97.5** | 76.09 |

Table 1: **Different classifiers can be best for different target distributions.** There is an inherent tradeoff between the robust and shortcut classifiers: ERM has higher total average accuracy, but JTT has higher worst-group accuracy.

| | Waterbirds | | CelebA | | MultiNLI | | MetaShift | |
|---|---|---|---|---|---|---|---|---|
| | Avg Acc | Avg Regret | Avg Acc | Avg Regret | Avg Acc | Avg Regret | Avg Acc | Avg Regret |
| ERM | 87.22 (0.15) | -5.94 (0.37) | 79.38 (0.44) | -16.00 (0.53) | 81.36 (0.35) | -0.8 (0.18) | 82.41 (0.74) | -3.05 (0.65) |
| JTT Liu et al. (2021) | 90.71 (0.22) | -2.5 (0.08) | 84.91 (0.11) | -9.92 (0.87) | 80.71 (0.24) | -1.43 (0.4) | – | – |
| LISA Yao et al. (2022) | 87.22 (0.14) | -5.94 (0.45) | 89.88 (0.63) | -6 (1.24) | – | – | 82.3 (0.45) | -3.19 (0.01) |
| Ensemble (logits) | 90.36 (0.11) | -2.81 (0.23) | 87.15 (0.33) | -7.89 (0.60) | **81.98 (0.32)** | **-0.22 (0.36)** | 82.56 (0.54) | -2.87 (0.32) |
| Ensemble (weights) | 50.00 (0) | -43.17 (0.29) | 50 (0) | -45.22 (0.65) | 81.58 (0.30) | -0.45 (0.48) | 79.57 (0.51) | -5.57 (0.41) |
| CoSMoS (input-dep) | 90.36 (0.15) | -2.81 (0.34) | 88.07 (0.32) | -7.03 (0.66) | 81.93 (0.28) | -0.26 (0.37) | 82.56 (0.54) | -2.87 (0.32) |
| CoSMoS (clusters) | **91.72 (0.19)** | **-0.74 (0.22)** | **90.96 (0.18)** | **-1.5 (0.95)** | 81.62 (0.30) | -0.58 (0.63) | **83.78 (0.4)** | **-1.49 (0.3)** |

Table 2: **Main results.** On each dataset, we evaluate accuracy on a wide range of representative test sets to reflect the wide range potential subpopulations that a model may need to be used for. We report average accuracy across all test sets as well as the average regret across the individual groups, with standard error across 5 seeds in the parentheses. We find that on the 3 image domains, CoSMoS (cluster) achieves significantly higher accuracy and lower regret than any of the individual classifiers or model aggregation through ensembling. On the text domain, CoSMoS (input-dep) matches Ensemble as the best-performing method.

**Evaluation Metrics.** In practice, we may want to use models that have been trained on some source distribution on a variety of potential test distributions with unknown subpopulation composition. Thus, the goal in our problem setting is to achieve high accuracy on *every* potential subgroup. Hence, when choosing our evaluation metrics, we want to capture each method's ability to handle a wide range of test distributions. We summarize performance on each dataset using two metrics: average accuracy across all test sets and average regret over the individual groups. The latter measures the average difference between our method's accuracy and the accuracy of the best-performing base classifier across each given individual subgroup. These metrics both evaluate each method's ability to consistently do well on a range of subpopulations.

## 5.2 MODELS TRAINED ON SOURCE DATA OFTEN PERFORM DIFFERENTLY ON DIFFERENT TEST DISTRIBUTIONS

We first aim to demonstrate how two different classifiers that are predictive on source data can perform differently on different target distributions. On the Waterbirds dataset, consider a robust feature learned by JTT (Liu et al., 2021) vs. an ERM classifier. In Tab. 1, as expected, the robust feature achieves the best worst-group accuracy. However, we find that the shortcut feature outperforms the robust feature on the two majority groups, indicating that this feature would achieve higher performance in a distribution skewed towards majority groups. In other words, there is no one best feature, and different features can be best for different target distributions. These observations justify CoSMoS: it can be beneficial to apply different features to different inputs, based on uncertainty. extract a diverse set of features that cover both causal and shortcut features, and adapt to different target distributions by interpolating between these learned features.

## 5.3 MAINTAINING LOW AVERAGE REGRET ACROSS EACH SUBGROUP

We investigate whether CoSMoS can consistently choose the best classifier for different subpopulations, and how this approach compares only using one model or other model aggregation methods. We perform a comprehensive experimental evaluation on the four datasets. Along with ERM and a robustness approach (either JTT or LISA), we evaluate four methods: (1) Ensemble (logits), which averages the logits of the three models, (2) Ensemble (weights), which averages the weights of the three models, similar to model soups (Wortsman et al., 2022a), (3) CoSMoS (input-dep), which selects different models based on confidence for different inputs but does not cluster the inputs before model selection, and (4) CoSMoS (cluster), which clusters the inputs before model selection. Experiments in Tab. 2 indicate that on our three image datasets, CoSMoS (cluster) significantly outperforms using any individual model or the ensembling approaches on average accuracy across all mixtures and individual groups as well as average regret over the groups, and CoSMoS (input-

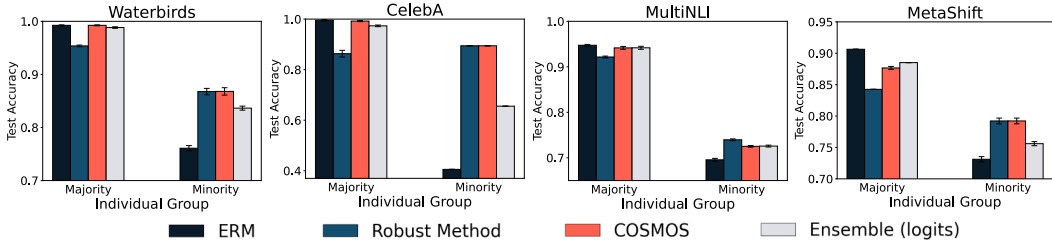

Figure 2: **Accuracies on mixture distributions.** We show the accuracy of each method on a range of test sets containing mixtures of majority and minority groups weighted differently. While each individual classifier has a subpopulation that it performs poorly on, COSMOS is able to leverage the strengths of each classifier to mitigate worst performance on a mixture.

Figure 3: **Accuracies on individual groups.** For each dataset, we show accuracies from two groups, one of which comes from the majority distribution and the other from the minority. We show that COSMOS is able to selectively assign different classifiers based on which is best for different groups.

dep) matches the best other method, Ensemble (logits), on the language domain. Clustering on the language domain does not improve performance because it does not effectively cluster inputs in the same subpopulations together, but COSMOS without clustering still performs well because it can select the best model for each input. In particular, these results show that we can achieve better performance than using a single invariant predictor, the strategy adopted by prior works by default, by using shortcut features when appropriate. This is because those shortcut features are valuable and are actually better than invariant predictors on some subpopulations, as shown in Table 1.

In Figure 2, we show the accuracy of each method on test mixtures of majority and minority groups on each dataset, and in Figure 3, we show the accuracy of each method on each individual group of the datasets. We observe that on the image domains, COSMOS consistently chooses the classifier that gives the best accuracy on the individual groups and on the mixtures of groups, and COSMOS without clustering does so on the language domain. For each dataset, for each individual classifier–whether it be a shortcut or invariant predictor, there is a subpopulation for which it is the worst classifier, and COSMOS is able to alleviate this issue, as there is no subpopulation on which it significantly underperforms. We also observe that COSMOS (cluster) is even able to achieve higher accuracy than the best individual classifier on some mixtures of groups–in these cases, it is best to use both classifiers on different inputs in the test set.

## 5.4 USING COSMOS FOR HYPERPARAMETER TUNING

Although our main motivation is to show how COSMOS can use multiple models improve performance across subpopulations, it can be used for other tasks where we want to choose the best model from a given set of models. In particular, we show that we can use COSMOS in order to do hyperparameter tuning without any target domain labels. For hyperparameter tuning, the typical practice is to use a target validation set. Such a set is implicitly assumed by prior works that study robustness or adaptation (Sagawa et al., 2020; Liu et al., 2021; Kirichenko et al., 2022), and such prior works have not effectively done hyperparameter tuning for a target domain without additional domain-specific information. In this experiment, shown in Figure 4, we take six JTT checkpoints for Waterbirds as our given models, which are trained using different hyperparameters (learning rate and weight decay), and we use COSMOS to choose the best model for each input. For each desired test distribution, the model that is most commonly chosen by COSMOS corresponds to the model with the highest accuracy on that test distribution, so we can take the model most commonly chosen as the model with the best hyperparameters for that target set. Thus, we can do hyperparameter

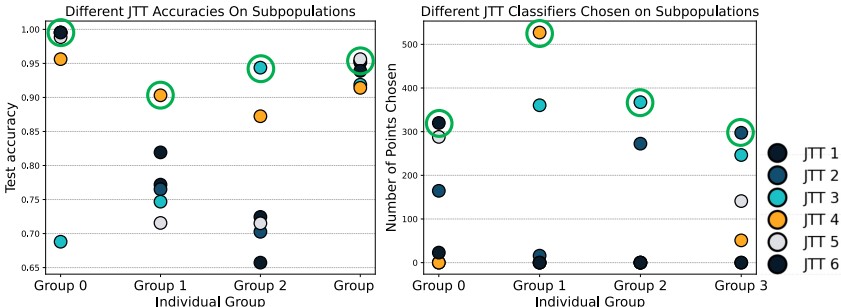

Figure 4: **Hyperparameter tuning with JTT models.** We are given 6 JTT (Liu et al., 2021) models trained on Waterbirds with different (learning rate, weight decay) hyperparameter combinations. We find that for each subpopulation, the classifier chosen for the most points by COSMOS corresponds to the classifier with the highest (or nearly highest) accuracy on that subpopulation. Thus, COSMOS is able to hyperparameter tuning for different target domains *without any target labels*.

tuning without any additional information from the desired test distribution, and to our knowledge, COSMOS is the first method that does so.

## 5.5 ABLATION ON CLUSTER SIZE

In this subsection, we ablate on the size of clusters used for COSMOS (cluster) on the image datasets.

In our above experiments, we cluster the test inputs into size(test set) / $N$ number of clusters, where $N = 50$. $N$ is the only additional hyperparameter that is used in COSMOS (cluster), but we do not tune $N$ above, and we find the value of $N = 50$ works for each image domain where clustering helps. In Table 3, we find that COSMOS (cluster) is not sensitive to the exact cluster size. However, there are small dips in performance with very small and very large cluster sizes, particularly in average regret. For the former, COSMOS (cluster) will become similar to COSMOS (input-dep) and the average confidence per cluster may be less reliable, and if the cluster size is too large, it may not be able to capture different mixtures of subpopulations that are in a test set and assign different classifiers to those different subpopulations.

| Cluster Size | Avg Acc | Avg Regret |
|---|---|---|
| 1 | 90.36 | -2.81 |
| 5 | 91.55 | -1.48 |
| 10 | 91.65 | -1.6 |
| 20 | 91.97 | -0.66 |
| 50 | 91.72 | -0.74 |
| 100 | 91.98 | -0.39 |
| 500 | 91.78 | -0.82 |

Table 3: **Ablation on cluster size.**

## 6 LIMITATIONS AND CONCLUSION

In this paper, we study the issue of spurious correlations in datasets and propose a framework called COnfidence-baSed MOdel Selection (COSMOS), to selectively employ appropriate classifiers for different inputs. Our method is based on the observation that different models have unique strengths on different regions of the input space, and using a single predictor may result in suboptimal performance. While prior methods typically focus on the average-case or worst-case subpopulation, COSMOS can achieve high accuracy across a range of subpopulations without sacrificing performance on any specific subpopulation. Our method does not require access to any labeled data from the target domain, any group annotations, or any model weights. We evaluate our method on four spurious correlation datasets each with multiple test distributions and find that COSMOS outperforms existing approaches, achieving 2-5% lower average regret across subgroups of the input space. Furthermore, our approach can be used for other use cases where it may be desirable to choose the best model for an unlabeled test set, including hyperparameter tuning. Despite the strengths of our framework, limitations remain. First, the performance relies on the capabilities of the base classifiers, and will not provide improvements if the base classifiers are all similar to one another. In addition, there are scenarios in which it is not be appropriate to use shortcut features; for example, in some applications, shortcut features may correspond to protected attributes, e.g. race, gender, age, etc. COSMOS should not be applied in such settings. Nevertheless, COSMOS is useful in other settings where we aim to achieve both high average accuracy and worst-group accuracy, and our results demonstrate the benefits of viewing shortcut and invariant classifiers on equal footing and selectively employing appropriate models for different inputs.

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

# A  DATASET DETAILS

Below, we describe our four datasets. For each, we calibrate the classifiers using held-out data from the source domain and evaluate on multiple different target distributions from either (a) mixing majority and minority groups or (b) each individual group. Our aim with constructing many test sets is to evaluate performance on a wide range of potential subpopulation shifts.

- **Waterbirds** (Sagawa et al., 2020). This dataset tasks the model with classifying images of birds as either a waterbird or landbird. The label is spurious correlated with the image background, which is either water or land. There are four individual groups; in our evaluation, we consider the two groups where the bird and background are correlated as majority groups and the others as minority groups. There are n = 4795 training examples and 56 in the smallest group (waterbirds on land).

- **CelebA** (Liu et al., 2015). The task is to classify the hair color of the person as blond or not blond, and this label is spuriously correlated with gender. There are four total groups; in our evaluation, we consider the two groups (blond, female) and (not blond, male) as majority groups and the two others as minority groups, and we additionally evaluate on all four groups individually. There are n = 162770 training examples in the CelebA dataset, with 1387 in the smallest group (blond-haired males).

- **MultiNLI** (Williams et al., 2018). The task is to classify whether the second sentence is entailed by, neutral with, or contradicts the first sentence in a pair of sentences. The label is spuriously correlated with the presence of negation words in the second sentence. There are 6 individual groups; we take (no neg, entailment), (neg, contradiction), (no neg, neutral) as majority groups and (neg, entailment), (no neg, contradiction), and (neg, neutral) as minority groups. There are n = 206175 examples in the training set, with 1521 examples in the smallest group (entailment with negations)

- **MetaShift** (Liang & Zou, 2022). This dataset is derived using the real-world images and natural heterogeneity of Visual Genome (Krishna et al., 2016). The model is tasked with identifying an image as a dog(shelf) or cat(shelf), given training data of cat(sofa), cat(bed), dog(cabinet), and dog(bed) domains. There are 2 individual groups, and we take the dog(shelf) images as one group and cat(shelf) as another to construct test domains. This task requires generalizing to novel data subpopulations not observed during training. The total size of training data is 400 images.

# B  ADDITIONAL EXPERIMENTAL RESULTS

## B.1  QUALITATIVE RESULTS

We cluster the test embeddings obtained using a ResNet50 pre-trained on ImageNet and a pre-trained BERT model for MultiNLI. Below, we show how clustering can allow us to estimate which datapoints have similar $z$, after which we can assign classifiers to the points using confidence with less noise. In Figure 5, on the Waterbirds dataset, although clustering does not recover the groups perfectly, it generally groups together datapoints in the same subgroup, and as pre-trained embeddings improve in the future, such clustering may also improve. We can leverage the strength of these pre-trained embeddings to better choose the appropriate classifier for different inputs.

## B.2  ADDITIONAL COMPARISONS

In Table 4 we include an additional comparison to random selection among the base classifiers. We see that COSMOS significantly outperforms this baseline on our 4 datasets on both average accuracy and average regret metrics, showing the benefits of our selection method.

| | Waterbirds | | CelebA | | MultiNLI | | MetaShift | |
|---|---|---|---|---|---|---|---|---|
| | Avg Acc | Avg Regret | Avg Acc | Avg Regret | Avg Acc | Avg Regret | Avg Acc | Avg Regret |
| Random | 88.35 | -4.02 | 84.47 | -9.72 | 80.9 | -1.42 | 79.39 | -3.4 |
| COSMOS (clusters) | 91.72 | -0.74 | 90.96 | -1.5 | 81.62 | -0.58 | 83.78 | -1.49 |

Table 4: **Comparison to Random Selection.** COSMOS significantly outperforms randomly selecting among the base classifiers.

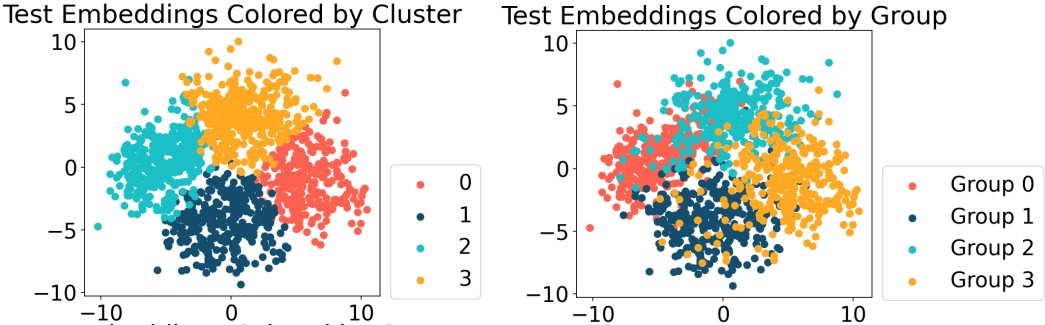

Figure 5: **Clustering recovers subgroups.** We plot the PCA projections on 2 dimensions of the test embeddings of the Waterbirds dataset, colored by cluster on the left and by group on the right. The points in a cluster chosen by K-means generally correspond to those in a single subgroup.

