# OpenReview forum: "Confidence-Based Model Selection: When to Take Shortcuts in Spurious Settings"
_ICLR.cc/2024/Conference — Submitted to ICLR 2024_

### Official Review · Reviewer_niM7 · 2023-10-24

**Soundness:** 2 fair
**Presentation:** 2 fair
**Contribution:** 2 fair
**Rating:** 3
**Confidence:** 3

**Summary:**

This paper concentrates on introducing a training method known as COnfidence-baSed MOdel Selection (COSMOS). The paper presents the COSMOS framework, which utilizes model confidence to adaptively select among models with varying strengths for distinct subpopulations. COSMOS does not necessitate target labels or group annotations, making it suitable for situations where obtaining such information is challenging. Nevertheless, this approach lacks adequate experimental analysis and falls short in the interpretability of its algorithm design.

**Strengths:**

- The paper presents a novel framework, COSMOS, which tackles the issue of distributional shift by selectively using suitable classifiers based on model confidence. COSMOS adaptively chooses models depending on their appropriateness for various inputs, taking into account both shortcut and invariant classifiers. The proposed approach does not depend on target labels or group annotations, making it applicable in situations where such information is inaccessible or challenging to obtain.

- This paper has a clear and rational motivation that advocates for treating shortcut and invariant classifiers equally, with both being experts in different regions of the input space.

- This paper provides a comprehensive formal definition of the problem, including the problem setting and formal intuition.

- To a certain degree, the algorithm in this paper demonstrates experimental results that the model can maintain satisfactory performance for majority groups while enhancing the performance of minority groups.

**Weaknesses:**

- The COSMOS framework assumes that test data is provided in a batch format, with multiple inputs available at once for model selection. However, in real-world situations, particularly in medical diagnosis where subpopulation shifts are common, test data may be received in a streaming manner, processing one sample at a time.

- The algorithm's design lacks interpretability. The analysis did not take into account the relationship between the algorithm's design and the use of shortcut and invariant features, nor did it explain why different classifiers can use various combinations of these two features instead of relying on the same shortcut features.

- COSMOS' performance depends on the abilities of the base classifiers. If the base classifiers are similar, COSMOS may not offer significant improvements.

- Another drawback related to numerous base classifiers is the need to train multiple base classifiers, each with potentially different architectures or training backbones. This increases the complexity and computational cost of the training process, as each base classifier must be trained and calibrated individually. Managing and optimizing multiple training pipelines can be difficult, particularly when working with large-scale datasets or complex models. In comparison to many existing methods that only require one base encoder (e.g., see [1, 2] and benchmarking methods in [3]), COSMOS displays increased training complexity.

- As the paper focuses exclusively on spurious correlations as a type of subpopulation shift, it neglects the wider variety of subpopulation shift types found in the literature. According to [3], subpopulation shifts can take many forms, such as attribute imbalance and class imbalance. Real-world datasets often exhibit multiple types of shifts at the same time, and the paper does not discuss how COSMOS would perform in these situations. As a result, the paper's limited scope undermines its generalizability and applicability to real-world datasets that may display different types of subpopulation shifts.

- The paper does not offer a comprehensive comparison with current state-of-the-art methods, making it challenging to evaluate COSMOS' relative performance and advantages compared to other techniques.

- Although the paper proposes considering metrics beyond worst-group accuracy (WGA), it only evaluates regret and does not acknowledge the tradeoffs between other essential metrics and their interactions. Recent research on subpopulation shifts [3, 4] has shown that metrics such as calibration error (ECE) or worst-case precision may conflict with WGA. As a result, it is crucial to carefully consider the limitations and potential trade-offs of alternative metrics when assessing the performance of the proposed COSMOS framework. How does COSMOS perform on those metrics?

- The ablation experiment is insufficient. The authors did not examine whether this advantage is due to the presence of TS. Moreover, if random selection or other selection methods are used among K classifiers, it is unclear whether the results will differ. It remains uncertain whether the advantage of the results is due to the integration of multiple classifiers.



_[1] Distributionally robust neural networks for group shifts: On the importance of regularization for worst-case generalization. ICLR 2020._

_[2] On feature learning in the presence of spurious correlations. NeurIPS 2022._

_[3] Change is Hard: A Closer Look at Subpopulation Shift. ICML 2023._

_[4] Simple data balancing achieves competitive worstgroup-accuracy. 2022._

**Questions:**

Please refer to the Weaknesses. In addition to the points raised above, I have the another question it seems the k should be k^i = \frac{D^i_T}{N}, since i denotes multiple target test sets.

---

> ### Author Response · Authors · 2023-11-17
>
> Thank you for your review. We address your comments below.
>
> > The COSMOS framework assumes that test data is provided in a batch format, with multiple inputs available at once for model selection. However, in real-world situations, particularly in medical diagnosis where subpopulation shifts are common, test data may be received in a streaming manner, processing one sample at a time.
>
> If the test data arrives in a streaming fashion, we can still apply COSMOS to each individual point that arrives, and we have a method, COSMOS (input-dep), that does this. Although we do not get the benefit of clustering, this method still outperforms the individual base classifiers.
>
> We note that assuming unlabeled test data in batch form is a realistic and powerful source of information, and is a common assumption in settings with distribution shifts, such as OOD generalization [1,2,3], test-time adaptation [4,5], and OOD detection [6,7]. There are a number of realistic settings where it is easy to collect unlabeled data from the target distribution whenever a machine learning model is deployed, for almost no additional cost [7,8], or to retrieve relevant unlabeled data from a large dataset based on similarity [9].
>
> [1] Zhang, Linfeng, et al. "Be your own teacher: Improve the performance of convolutional neural networks via self distillation." ICCV 2019
>
> [2] Xie, Qizhe, et al. "Self-training with noisy student improves imagenet classification." CVPR 2020
>
> [3] Sohn, Kihyuk, et al. "Fixmatch: Simplifying semi-supervised learning with consistency and confidence." NeurIPS 2020
>
> [4] Wang, Dequan, et al. "Tent: Fully test-time adaptation by entropy minimization." ICLR 2021
>
> [5] Zhang, Marvin, Sergey Levine, and Chelsea Finn. "Memo: Test time robustness via adaptation and augmentation." NeurIPS 2022
>
> [6] Katz-Samuels, Julian, et al. "Training ood detectors in their natural habitats." ICML 2022
>
> [7] Tifrea, Alexandru, Eric Stavarache, and Fanny Yang. "Semi-supervised novelty detection using ensembles with regularized disagreement." Uncertainty in Artificial Intelligence. PMLR, 2022.
>
> [8] Sagawa, Shiori, et al. "Extending the WILDS benchmark for unsupervised adaptation." ICLR 2022
>
> [9] Schuhmann, Christoph, et al. "Laion-5b: An open large-scale dataset for training next generation image-text models." NeurIPS 2022
>
>
> > The algorithm's design lacks interpretability. The analysis did not take into account the relationship between the algorithm's design and the use of shortcut and invariant features, nor did it explain why different classifiers can use various combinations of these two features instead of relying on the same shortcut features.
>
> The intuition behind why our method utilizes both shortcut and invariant features is described in Section 4.1. In summary, the inference problem can be formulated as smoothed entropy minimization, which motivates our use of confidence in choosing which classifier to use. Shortcut and invariant features naturally have different confidences on different inputs as a result of how they are trained and this is also reflected in how they have different accuracies on different groups of inputs (as shown in Table 1). Thus, by leveraging these different confidences, our method is able to utilize different combinations of both shortcut and invariant features.
>
>
> > COSMOS' performance depends on the abilities of the base classifiers. If the base classifiers are similar, COSMOS may not offer significant improvements.
>
> COSMOS assumes access to a diverse set of base classifiers, and the framework benefits from diversity in the prediction strategies of the base models. Such diverse models are often available in the distribution shift settings we consider. In the extreme worst case of all models being equal, COSMOS does as well as the single model.
>
> > Another drawback related to numerous base classifiers is the need to train multiple base classifiers, each with potentially different architectures or training backbones. This increases the complexity and computational cost of the training process, as each base classifier must be trained and calibrated individually...
>
> We do use different base classifiers, which may require additional compute over training a single model, but we find that this is worth it for the additional improvement in performance that we get from maintaining high majority and minority group accuracies. Furthermore, other methods like Ensembling do also train and use multiple models and we find in our experiments that we are able to leverage the models more effectively for better performance.

---

> > ### Author Response · Authors · 2023-11-17
> >
> > > As the paper focuses exclusively on spurious correlations as a type of subpopulation shift, it neglects the wider variety of subpopulation shift types found in the literature...
> >
> > We focus on datasets with spurious correlations because our aim is to highlight the benefits of leveraging the strengths of both shortcut and debiased classifiers, which are commonly trained for datasets with spurious correlations. Our method is designed to benefit particularly when given shifts in the frequencies of data subpopulations, as different classifiers often perform better on different subpopulations in this setting. However, the method can also be used to generalize to novel data subpopulations that were not observed during training and does so in the MetaShift evaluation.
> >
> > > The paper does not offer a comprehensive comparison with current state-of-the-art methods, making it challenging to evaluate COSMOS' relative performance and advantages compared to other techniques.
> >
> > We compare to state-of-the-art classifiers and methods that use multiple classifiers (e.g. ensembling variants). If you have any recommendations for other specific methods that would be informative to compare against, we would be happy to consider including them.
> >
> > > Although the paper proposes considering metrics beyond worst-group accuracy (WGA), it only evaluates regret and does not acknowledge the tradeoffs between other essential metrics and their interactions. Recent research on subpopulation shifts [3, 4] has shown that metrics such as calibration error (ECE) or worst-case precision may conflict with WGA. As a result, it is crucial to carefully consider the limitations and potential trade-offs of alternative metrics when assessing the performance of the proposed COSMOS framework. How does COSMOS perform on those metrics?
> >
> > Can you specify which metrics you are referring to? We consider performance on a wide range of test distributions, including individual groups; thus, our evaluation includes worst-group performance along with other metrics.
> >
> > > The ablation experiment is insufficient. The authors did not examine whether this advantage is due to the presence of TS. Moreover, if random selection or other selection methods are used among K classifiers, it is unclear whether the results will differ. It remains uncertain whether the advantage of the results is due to the integration of multiple classifiers.
> >
> > Temperature scaling is a part of COSMOS and is key to ensuring that the classifiers are similarly calibrated. Our main results, shown in Table 2, show that on all four domains, using COSMOS achieves significantly higher accuracy and lower regret than any of the individual classifiers, demonstrating how multiple classifiers can be used to obtain performance gains on a wide range of potential subpopulations.
> >
> > Below, we include an additional comparison to random selection among K classifiers. We see that COSMOS significantly outperforms this baseline on our 4 datasets on both average accuracy and average regret metrics, showing the benefits of our selection method.
> >
> > |                   | Waterbirds |          | CelebA   |          | MultiNLI |          | MetaShift |          |
> > |:-----------------:|:----------:|:--------:|:-------:|:--------:|:--------:|:--------:|:---------:|:--------:|
> > |                   |   Avg Acc  | Avg Regret | Avg Acc | Avg Regret | Avg Acc  | Avg Regret | Avg Acc   | Avg Regret |
> > |       Random      |    88.35   |    -4.02   |  84.47  |    -9.72   |   80.9   |    -1.42   |   79.39   |    -3.4    |
> > | COSMOS (clusters) |    91.72   |    -0.74   |  90.96  |    -1.5    |   81.62  |    -0.58   |   83.78   |    -1.49   |
> >
> >
> > > I have the another question it seems the k should be k^i = \frac{D^i_T}{N}, since i denotes multiple target test sets
> >
> > Thanks for catching this typo. We have fixed this in the updated pdf.
> >
> > We hope that our response has addressed all your questions and concerns. We kindly ask you to let us know if you have any remaining concerns, and - if we have answered your questions - to reevaluate your score.

---

> > > ### Comment · Reviewer_niM7 · 2023-11-20
> > > **Metric**
> > >
> > > Thank you for your rebuttal, the calibration metric ECE please refer to [A].
> > >
> > > [A] Guo, Chuan, et al. "On calibration of modern neural networks." International conference on machine learning. PMLR, 2017.

---

> > > > ### Author Response · Authors · 2023-11-21
> > > >
> > > > We find that in our experimental settings, COSMOS tends to be robust to noisy confidence estimates, as aided by the clustering mechanism. The focus of our work is not on the interplay between calibration and other metrics. Instead, our focus is on the empirical benefits of utilizing both causal and shortcut features for improved performance in settings with spurious correlations. We leave in-depth studies on the nature of calibration and this interplay for separate work.
> > > >
> > > > COSMOS is a method that dynamically chooses among base classifiers, so its calibration error aligns with that of the base classifiers. In order to calibrate the base classifiers, we use temperature scaling with the following procedure: we tune over different temperatures, taking the temperature that gives the lowest expected calibration error (ECE) for one classifier and choosing the temperature for the other classifiers that gives the ECE closest to the one for the first classifier. This procedure allows the classifiers to be similarly calibrated and is described formally in Section 4.2.
> > > >
> > > > Again, thank you for your review. Please let us know if you have any remaining questions. We are open to discussion and would be happy to provide further clarifications.

---

> > > > > ### Author Response · Authors · 2023-11-22
> > > > > **Following up**
> > > > >
> > > > > Thanks again for your review. We wanted to follow up again to make sure that your concerns are being properly addressed. Please let us know if you have additional questions. if all your concerns have been resolved, we would greatly appreciate it if you could reconsider and adjust your rating and evaluation of our work.

---

### Official Review · Reviewer_RGsk · 2023-10-31

**Soundness:** 2 fair
**Presentation:** 2 fair
**Contribution:** 2 fair
**Rating:** 5
**Confidence:** 4

**Summary:**

The paper discusses the challenges and solutions in machine learning related to feature learning, model robustness, and calibration. It highlights the importance of identifying shortcut features, which are often ignored in favor of robust predictors. The authors propose a technique called COnfidence-baSed MOdel Selection (COSMOS) that uses model confidence to guide model selection without the need for target labels or group annotations. They show that COSMOS outperforms other methods on datasets with distributional shift. Additionally, the paper introduces a fewshot recalibration approach to improve model calibration for specific data slices, demonstrating its effectiveness in various downstream tasks.

**Strengths:**

1) The paper introduces a unique approach (COSMOS) for model selection based on model confidence, which does not rely on target labels or group annotations, addressing a common challenge in machine learning.
2) The paper demonstrates that COSMOS performs better than other model aggregation methods on datasets with distributional shift, achieving lower regret across subpopulations.
3) The approach is general to be applied to a wide range of models.

**Weaknesses:**

1) There exists gap between the formal intuition and practical approach. Some assumptions are strict to stand in practice. The rationality of theories and the gap between theories and methods need to be addressed. Otherwise, we have no way of knowing the scope of the method.
2) More methods, as well as some SOTA , should be considered in experiment.

**Questions:**

nan

**Details Of Ethics Concerns:**

nan

---

> ### Author Response · Authors · 2023-11-17
>
> Thanks for your review. We'd like to ask for some clarification on the weaknesses you indicated.
>
> >There exists gap between the formal intuition and practical approach. Some assumptions are strict to stand in practice. The rationality of theories and the gap between theories and methods need to be addressed. Otherwise, we have no way of knowing the scope of the method.
>
> We apologize for any potential confusion here. Can you explain what you find to be the gap between our theory and method so that we can clarify this?
>
> > More methods, as well as some SOTA, should be considered in experiment.
>
> We already compare to state-of-the-art classifiers and methods that use multiple classifiers (e.g. ensembling variants). If you have any recommendations for other specific methods that would be informative to compare against, we would be happy to consider including them.
>
> We kindly ask you to expand on your concerns if they remain, and to reevaluate your score if they are resolved.

---

> > ### Author Response · Authors · 2023-11-21
> > **Checking in**
> >
> > We wanted to follow up on your review and our response. We are open to discussion if you have any additional questions or concerns, and if not, we kindly ask you to reevaluate your score and assessment of our work.

---

> > > ### Author Response · Authors · 2023-11-22
> > > **Following up**
> > >
> > > Thanks again for your review. We wanted to follow up again to make sure that your concerns are being properly addressed. Please let us know if you have additional questions. If all your concerns have been resolved, we would greatly appreciate it if you could reconsider and adjust your rating and evaluation of our work.

---

### Official Review · Reviewer_4daw · 2023-10-31

**Soundness:** 4 excellent
**Presentation:** 3 good
**Contribution:** 4 excellent
**Rating:** 8
**Confidence:** 4

**Summary:**

This paper studies the problem of confidence-based model selection in an effort to address the issue of distribution shifts in the testing phase by equally considering invariant and shortcut features. Given multiple base classifiers trained on the source dataset, the COSMOS algorithm is proposed that (1) first clusters test examples in K clusters, and (2) then uses a confidence score to select one out of the base classifiers to perform classification of the examples for each cluster. The performance of the proposed algorithm is evaluated on 4 datasets  and compared with methods that use only invariant features, only shortcut features and ensemble methods.

**Strengths:**

+ The problem of addressing distribution shifts in the testing dataset is addressed when spurious correlations are present.
+ The idea of using different models for different inputs is very neat.
+ The proposed algorithm is simple and very intuitive and has a nice formal intuition.
+ The performance of the proposed algorithm is validated using 4 datasets, illustrating its superior performance compared to existing works.
+ The proposed algorithm improves classification performance in real-world scenarios that are prevalent with distribution shifts and spurious correlations.
+ The paper is in general well-written and makes it easy for the reader to understand both the problem statement and the solution.

**Weaknesses:**

- I believe that the solution presented in the paper relates also to the problem of dynamic or instance-wise classifier selection, where the goal is to select the best classifier to use during testing for each test example. The related work section does not seem to include any relevant work in this area. Some example references follow:

 (1) R. M. Cruz, R. Sabourin, and G. D. Cavalcanti, Dynamic classifier selection: Recent advances and perspectives, Information Fusion, vol. 41, pp. 195–216, 2018.

(2) M. Sellmann and T. Shah, Cost-sensitive hierarchical clustering for dynamic classifier selection, arXiv preprint arXiv:2012.09608, 2020.

(3) R. M. O. Cruz, L. G. Hafemann, R. Sabourin, and G. D. C. Cavalcanti, Deslib: A dynamic ensemble selection library in python, Journal of Machine Learning Research, vol. 21, no. 8, pp. 1–5, 2020.

(4) S. P. Ekanayake, D. Zois and C. Chelmis, Sequential Datum-Wise Joint Feature Selection and Classification in the Presence of External Classifier, IEEE International Conference on Acoustics, Speech and Signal Processing (ICASSP), Rhodes Island, Greece, 2023, pp. 1-5, doi: 10.1109/ICASSP49357.2023.10097057.

**Questions:**

(1) It would be great if the authors discuss how the proposed method differs from the problem of dynamic or instance-wise classifier selection, and depending on the relevance, they will consider extending their related work section accordingly.

(2) I am a little bit confused about the relationship between the index variable of the subpopulation and the label. Initially, I thought that the test set could be split into subpopulations based on the possible values of the labels. However, as I continued reading, it seems that subpopulations are not necessarily constructed based on the possible values of the labels. In this case, what is the meaning of subgroups and how do you justify this?

(3) Can you explain what is the meaning of the invariance assumptions in Sec. 3?

Minor:
(a) I believe there is a small typo in notation. Namely, shouldn't p_{T_i} be p_{T^i} in Sec. 3 or am I confused?
(b) In pg. 5, dist(.) should be properly defined as a divergence measure.
(c) The statistics of the datasets (e.g., number of instances, features, etc) are not reported.

---

> ### Author Response · Authors · 2023-11-17
>
> We thank you for your thoughtful review and appreciate your acknowledgment of many strengths of our work. If you have any remaining questions, please let us know.
>
> > It would be great if the authors discuss how the proposed method differs from the problem of dynamic or instance-wise classifier selection, and depending on the relevance, they will consider extending their related work section accordingly.
>
> Thank you for bringing this up and the links to references. We have added these and some discussion to the related work section accordingly. Our method is an example of dynamic selection (DS) from multiple classifiers; we propose a distinct selection criteria, combining both clustering and confidence strategies in order to enhance performance on subpopulation shifts.
>
> > I am a little bit confused about the relationship between the index variable of the subpopulation and the label. Initially, I thought that the test set could be split into subpopulations based on the possible values of the labels. However, as I continued reading, it seems that subpopulations are not necessarily constructed based on the possible values of the labels. In this case, what is the meaning of subgroups and how do you justify this?
>
> Thanks for bringing this point up, and we apologize for any confusion. In subpopulation shifts that occur as a result of spurious correlations, we follow prior literature, e.g. Group DRO, JTT, etc., and consider different combinations of (spurious attribute, label) as a group. For example, for the Waterbirds dataset, where background type spuriously correlates with the label of bird type, we consider 4 groups total, 2 of them majority: (land background, landbird), (water background, waterbird), and 2 of them minority (water background, landbird), (land background, waterbird).
>
> > Can you explain what is the meaning of the invariance assumptions in Sec. 3?
>
> We apologize for any confusion here. The invariance assumptions simply define the scope of the distribution shifts that we consider. We are specifically interested in subpopulation shifts where the main difference between the source distribution and target distributions is the proportion of subpopulations and other aspects between the distributions remain the same.
>
> > Minor: (a) I believe there is a small typo in notation. Namely, shouldn't p_{T_i} be p_{T^i} in Sec. 3 or am I confused? (b) In pg. 5, dist(.) should be properly defined as a divergence measure. (c) The statistics of the datasets (e.g., number of instances, features, etc) are not reported.
>
> Thank you for your careful reading of our paper. We have fixed these points in the revised pdf.

---

> > ### Comment · Reviewer_4daw · 2023-11-21
> >
> > Thanks for clarifying my concerns. Now that I have gained a better understanding of the invariance assumption, I am thinking that the proposed approach must related to research focusing on imbalanced datasets since you are changing the proportion for subpopulations. In that sense, I think it is wise to discuss how the proposed approach differs from prior work in the area.

---

### Meta-Review · Area_Chair_LHwp · 2023-12-11

**Metareview:**

This paper proposes a method to perform classification to limit failures of out of distribution data.  Given different base classifiers trained on the source dataset (one for each subgroup), the COSMOS algorithm clusters test examples into K clusters, and for each example uses a confidence score to use one of the base classifiers to perform classification. The intuition herein is based on using spurious features to identify the most confident classifier to use to make a prediction. The reviewers found the idea simple, and easy to evaluate but were unable to reach consensus on the work.

Two of the reviewers raised concerns about the lack of theory on when the method was expected to work as well as unclear intuition on why it was proposed (the authors do provide a nice response to the latter question). Reviewers also expressed concerns about the lack of comparisons to (relevant) baselines and about a lack of ablation studies -- while the authors provide responses to most questions in the rebuttal there was no consensus at the end of the review period.

Personally, I found the idea interesting and that it merits further investigation but agree the manuscript had room for improvement. If i had to pinpoint one key concern raised it was that in the absense of theory on when COSMOS is expected to work the reviewers were hoping to see larger experimentation on the sensitivity of the method to different choices in calibrating the classifier, a study of how changing the complexity of the base classifier affects COSMOS, and an ablation depicting how it may be used in a streaming fashion. I encourage the authors to incorporate the above as revisions.

In addition to the reviewers' comments (with respect to related work tackling this topic) I recommend the authors contextualize/compare against [1] in revisions; [1] also tackles the issue of robust predictions (which is represented via invariant predictors) seeing a reduction in prediction accuracy in subpopulations and provides asymptotic guarantees around the optimality of the predictor. I think it is a natural baseline to contrast COSMOS against.

[1] Spuriosity Didn't Kill the Classifier: Using Invariant Predictions to Harness Spurious Features, Eastwood et. al, NeurIPS 2023 (https://arxiv.org/abs/2307.09933)

**Justification For Why Not Higher Score:**

Justification provided in review.

**Justification For Why Not Lower Score:**

N/A

---

### Decision · Program_Chairs · 2024-01-16

Reject